# 1 Mapping global onshore wind turbines using multi-source remote

| 2  | se  | nsing images and hybrid learning approaches                                                                      |
|----|-----|------------------------------------------------------------------------------------------------------------------|
| 3  | Sh  | ujun Li <sup>1,6</sup> , Jianchuan Qi <sup>2,3,4,*</sup> , Yongze Song <sup>5</sup> , Peng Wang <sup>1,6,*</sup> |
| 4  |     |                                                                                                                  |
| 5  | 1.  | State Key Laboratory for Ecological Security of Regions and Cities, Institute of Urban                           |
| 6  |     | Environment, Chinese Academy of Sciences, Xiamen 361021, China                                                   |
| 7  | 2.  | School of Environment, Tsinghua University, Beijing, 100084, China                                               |
| 8  | 3.  | Institute for Carbon Neutrality, Tsinghua University, Beijing, 100084, China                                     |
| 9  | 4.  | TianGong Think Tank, Research Institute for Environmental Innovation (Suzhou) Tsinghua,                          |
| 10 |     | 215163, China                                                                                                    |
| 11 | 5.  | School of Design and the Built Environment, Curtin University, Perth, Australia                                  |
| 12 | 6.  | University of Chinese Academy of Sciences, Beijing, 100049, China                                                |
| 13 | * ] | To whom correspondence may be addressed.                                                                         |
| 14 | Со  | rrespondence to: pwang@iue.ac.cn, jcqi@tsinghua.edu.cn                                                           |
| 15 |     |                                                                                                                  |
| 16 |     |                                                                                                                  |
| 17 |     |                                                                                                                  |
| 18 |     |                                                                                                                  |
| 19 |     |                                                                                                                  |
| 20 |     |                                                                                                                  |
| 21 |     |                                                                                                                  |
| 22 |     |                                                                                                                  |
| 23 |     |                                                                                                                  |
| 24 |     |                                                                                                                  |
| 25 |     |                                                                                                                  |
| 26 |     |                                                                                                                  |

58

**Abstract.** Wind power serves as a vital zero-carbon alternative to fossil fuels for climate change mitigation. Nevertheless, the vast expansion of wind turbine installation requires 31 32 extensive terrestrial resources, raising wide concerns regarding land use competition and 33 ecological impacts. Quantifying these effects necessitates near real-time geospatial data on 34 wind turbine placement and density. However, current methods remain inadequate monitoring 35 for the fast-growing wind turbine deployment. Here, we developed an integrated framework 36 that combines OpenStreetMap (OSM) data with multi-source remote sensing images (Google 37 Earth and Sentinel-1/2) and deep learning and traditional machine learning models (ResNet-18 and Random Forest) to map global onshore wind turbines. Our models achieve validation 38 39 accuracy >97% while enabling cost-effective, timely updates of global onshore wind turbines. 40 Eventually, we established a geographical dataset covering a total of 379,595 wind turbines globally by 2024. This dataset represents a tenfold expansion over currently available global 41 42 wind turbine inventories as of 2020. In addition, we found that 80% wind turbines are situated 43 on cropland and grassland, followed by forest and bare ground. This dataset facilitates essential 44 studies on renewable energy land management, ecological impact analysis, and data-driven 45 energy transition policies. The codes and dataset of the global onshore wind turbines is available 46 at Zenodo link: https://doi.org/10.5281/zenodo.17217523 (Shujun et al., 2025). 47 48 49 50 51 52 53 54 55 56

## 1 Introduction

60 Wind energy will increase substantially over the coming decades to meet clean energy targets 61 (Mckenna et al., 2025). Under the 1.5 °C scenario, global installed wind power capacity is projected to reach nearly 10,300 GW by 2050, with onshore wind comprising 75% of total 62 63 installations (Raimi et al., 2023). Compared to other energy technologies, wind power exhibits 64 relatively low land use efficiency when accounting for turbine spacing requirements (Dai et al., 65 2024). Accordingly, meeting future deployment targets will necessitate substantial land 66 allocations, raising pressing concerns about land-use conversion and biodiversity loss that 67 demand urgent attention (Kati et al., 2021; Rinne et al., 2018). However, detailed geospatial 68 data at the facility level is particularly required for the quantification of these impacts 69 (Kruitwagen et al., 2021). 70 Indeed, asset-level data and facility arrangement are essential for power generation nowcasting 71 and forecasting, as well as for decision-making by grid operators and energy stakeholders 72 (Calvert et al., 2013; Tavakkoli et al., 2021). For instance, geospatial analysis of historical 73 placements can inform turbine siting decisions by revealing both human and environmental 74 landscape factors (Roddis et al., 2018). Previous research confirmed that substantial positional 75 errors exist in the current available wind facility records, especially pronounced in high-growth 76 renewable energy markets (Cerri et al., 2024; Effenberger and Ludwig, 2022). A timely 77 geospatial data set is critically needed to maintain accurate records of wind energy 78 infrastructure, given its unprecedented growth rate. The dataset could also support data-driven 79 metrics for Sustainable Development Goals (SDGs) (Mishra et al., 2024), including SDG 7 80 (Affordable and Clean Energy), SDG 13 (Climate Action), and SDG 15 (Life on Land). 81 Despite the demonstrated importance of location data, only a few spatially explicit datasets are 82 publicly available. At the global scale, there is a geospatial wind turbine dataset for 2020 is 83 introduced (Dunnett et al., 2020), but its update mechanism depends entirely on OpenStreetMap 84 (OSM), a crowdsourced data derived from heterogeneous contributors that could introduce 85 significant uncertainty. Meanwhile, while multiple frameworks exist for updating global 86 offshore wind turbine data (Hoeser et al., 2022; Zhang et al., 2021), onshore turbine updating

87 methods remain underdeveloped due to their greater spatial distribution and environmental variability. Recently, Microsoft and Planet's Global Renewables Watch platform employs deep 88 89 learning for global wind and solar monitoring (Robinson et al., 2025), but demands massive 90 computing resources and provides only web-based queries without editable datasets. At the 91 national level, there are geospatial datasets for the United States (Rand et al., 2020), Germany 92 (Manske et al., 2022), and Italy (Smeraldo et al., 2020). However, inconsistent data collection 93 method across datasets with delays in update frequencies could hinder their systematic 94 comparability. Currently, the research community lacks both a unified methodology and 95 accessible datasets for tracking worldwide onshore wind turbine deployments. 96 To address these gaps, our study presents a hybrid framework combining deep learning and 97 traditional machine learning framework for updating global onshore wind turbine data. By 98 integrating multi-source remote sensing data (Google Earth high-resolution images, Sentinel-99 1, and Sentinel-2), our workflow systematically detects and validates global onshore wind 100 turbines to generate a 2024 geodatabase. With OSM turbine locations as initial inputs, the two-101 stage locating process involves: (1) training a deep learning classifier (ResNet-18) on Google 102 high-resolution images to identify and correct erroneous OSM records, followed by (2) 103 detecting omitted turbines with Sentinel-1/2 spectral features and a Random Forest model 104 trained on Google Earth Engine (GEE). Additionally, we examined worldwide land use 105 characteristics of wind turbine sites and their national distribution patterns to assess current 106 wind energy spatial utilization. Our study delivers comprehensive monitoring tools and datasets 107 essential for tracking wind energy growth, enabling data-driven policy decisions to advance 108 sustainable wind power development worldwide.

109

116117

123124

#### 2 Materials and methods

#### 2.1 Framework

The proposed framework combines OSM's crowdsourced geospatial data with a two-stage deep learning/traditional machine learning pipeline (Figure 1) to locate a comprehensive global wind turbine location dataset for 2024. The first part involves utilizing OSM turbine coordinates to extract high-resolution Google Earth images, then training a ResNet-18 convolutional neural network to classify and flag erroneous wind turbines in the OSM dataset. The second part employs confirmed turbine locations to train a Random Forest classifier for potential omitted wind turbines using Sentinel-1/2 features at GEE, conbining with validation through our pretrained ResNet-18 model applied to Google high-resolution images of the potential points. The integrated output merges error-corrected OSM data with supplemented wind turbine omissions, generating an enhanced global dataset that demonstrates improved spatial accuracy and comprehensive operational wind turbine coverage.

Figure 1. Framework for mapping global onshore wind turbines. Where the WT represents

wind turbines, OSM represents OpenStreetMap.

## 2.2 Two-phase approach for global onshore wind turbine mapping

# 127 2.2.1 Filtering of erroneous data with deep learning model

We obtained the baseline OSM 2024 wind turbine dataset through QGIS using the Overpass

API plugin with the query parameter: '["generator: source"="wind"]'. We refined the dataset by applying OSM land polygons (<a href="https://osmdata.openstreetmap.de/data/land-polygons.html">https://osmdata.openstreetmap.de/data/land-polygons.html</a>), resulting in a preliminary global inventory of 377,154 geolocated onshore turbines with complete metadata records. Given OSM's crowdsourced feature due to unverified contributors, the extracted turbine locations serve as initial references that demand thorough validation. Subsequent analysis must systematically address both commission errors (false positives) and omission errors (omitted turbines) through technical verification.

Figure 2. Spatial distribution of training samples (green points).

Based on the OSM-derived turbine coordinates, we created  $500m \times 500m$  extraction zones (QGIS Buffer Tool) to acquire high-resolution Google Earth images. This conservative spatial buffer accounts for maximum turbine diameters ( $\leq 200m$ ) while guaranteeing full rotor coverage (Muller et al., 2024). The image tiles were resized to a standardized  $256 \times 256$  pixel format, maintaining optimal input dimensions for our ResNet-18 architecture while retaining essential turbine characteristics. For model construction, we employed a strategically sampled 10% subset (37,285 images) from the complete dataset, which balancing representativeness with computational constraints during training. The spatial distribution of sampled turbine points exhibits balanced representation across global regions in Figure 2, confirming our stratified random sampling approach effectively maintained geographic diversity. This subset was manually annotated with labels for 'turbines' and 'non-turbines'. The labeled data was then split into training (60%, 22,372 images) and testing sets (20%, 7,457 images) validation sets (20%, 7,456 images) for our OSM error classification model. Representative samples of the buffered turbine images are displayed in Figure 3. The visual data reveal that turbines are

- distributed across diverse landscapes, including grasslands, bare land, cropland, and forests,
- with occasional installations near water bodies and built environments.

© Google Earth 2024

Figure 3. Different land types of onshore wind turbines in Google Earth images.

For automated classification of OSM wind turbine data, we employed the ResNet-18 architecture (He et al., 2016), leveraging its demonstrated image classification capabilities while ensuring computational efficiency for geospatial applications at scale. Our optimized ResNet-18 model processed all 339,869 candidate images, identifying 291,501 confirmed turbine locations (85.8% positive rate) while classifying 48,368 as non-turbine cases (14.2%). All negative classifications underwent rigorous cross-platform verification using Google Earth, Bing Maps, and Sentinel-2 images, enabling the removal of inaccurate OSM entries. These validated results were then integrated with the training data to generate an enhanced global turbine dataset with improved accuracy. The dataset and codes for training model are available

at Zenodo website: https://doi.org/10.5281/zenodo.17217523 (Shujun et al., 2025).

#### 2.2.2 Supplementing omitted data with traditional machine learning model

Based on the deep learning-classified OSM turbine dataset, we developed an optimized Random Forest model for comprehensive omission detection (Rigatti, 2017). The Random Forest model was trained on GEE using verified wind turbine locations from OSM, alongside globally sampled negative samples. We trained the Random Forest model with 10,000 globally distributed wind turbine locations (positive samples) and 20,000 non-turbine points (negative samples), and applied a 30m spatial buffer to negative samples to ensure characteristic representation. The dataset was then split into 70% training and 30% testing sets as illustrated in Figure 4.

**Figure 4.** Spatial distribution of train and test datasets for the Random Forest model. The green ones represent the points selected for model training, and the orange ones represent the points selected for model testing.

In addition to the original spectral bands from Sentinel-1 and Sentinel-2, we incorporated the Normalized Difference Vegetation Index (NDVI) (Huang et al., 2021) and the Normalized Difference Built-up Index (NDBI) (Zha et al., 2003) to enhance the differentiation between wind turbines and their background features. For comprehensive feature characterization, we implemented a random sampling strategy across 10,000 turbine locations, while covering all major wind development regions for reliable spectral analysis. Figure 5 presents seven selected spectral feature value distribution of wind turbines, revealing distinct characteristic ranges for turbine signatures across different sensor bands. This demonstrates the effectiveness of different band features in wind turbine classification. To reduce the computational load of the Random Forest model, we excluded the 800-m buffer area of already validated wind turbines and then

194 195

- defined upper and lower threshold boundaries to filter out non-turbine areas during the initial processing stage. These thresholds include Sentinel-2's B2 [0, 0.3], B3 [0, 0.3], B4 [0, 0.3],
- NDVI [0, 0.7], NDBI [0, 0.7], and Sentinel-1's VV [-18, 18] and VH [-25, 1].

Figure 5. Feature value distribution of randomly selected wind turbine samples.

Figure 6. Feature importance ranking for building a Random Forest classification model.

The final dataset incorporated 19-dimensional feature data for each sample point, which was

utilized for training the model to detect omitted wind turbine points. Our feature importance ranking of the 19-dimensional feature space (Figure 6) revealed that Sentinel-1's VV and VH polarization bands are particularly effective for identifying the wind turbines. This could contribute to the band's high sensitivity to vertical metallic structures such as turbine towers, as these act as corner reflectors that generate distinct bright signatures in SAR imagery. The Sentinel-2's B12 and B2 bands also show strong response to turbine structures, which enhances their contrast against natural backgrounds like vegetation, soil, and water.

#### 2.3 Classification accuracy assessment of models

We evaluated the performance of both deep learning and traditional machine learning models using standard classification metrics computed from confusion matrices, namely precision, recall, and F1-score, as shown in Eq. (1)-(3), as based on an independent validation set (Congalton, 1991; Goutte and Gaussier, 2005). Producer's accuracy (recall) quantifies the proportion of actual turbine locations correctly detected, while user's accuracy (precision) represents the fraction of predicted turbines that are true positives. Where the precision equals the number of true positives (TP) divided by the sum of true positives (FP). Where the recall equals the number of true positives (TP) divided by the sum of true positives (TP) and false negatives (FN). The F1-score harmonizes these metrics, providing particularly valuable evaluation for imbalanced turbine detection scenarios where background features significantly outnumber target objects.

$$Precision = \frac{TP}{TP + FP} \quad (1)$$
$$Recall = \frac{TP}{TP + FN} \quad (2)$$
$$F1 - score = 2 \times \frac{Precision \times Recall}{Precision + Recall} \quad (3)$$

#### 2.4 Land use occupation analysis of onshore wind turbines

This study utilizes ESRI's 2023 Land Use/Land Cover (LULC) dataset (Karra et al., 2021), derived from ESA Sentinel-2 images at 10-meter resolution, for analyzing land use characteristics surrounding onshore wind turbines. The LULC composite maps integrate annual predictions for nine defined categories, namely cropland, rangeland, forest, built-up areas, bare ground, water bodies, flooded vegetation, snow/ice cover, and cloud cover. By conducting

spatial overlay analysis between our finalized global onshore wind turbine dataset and the LULC classification within GEE, we characterized land occupation patterns through the extraction of underlying land use types at turbine sites. Additionally, we evaluated wind turbine land use impacts by conducting 800-meter buffer analyses around turbine locations (Dunnett et al., 2020), and converting the results to raster format for comprehensive spatial assessment.

## 3 Results

#### 3.1 Evaluation results

Figure 7a displays the deep learning model's performance for onshore wind turbine error filtering, achieving exceptional precision (99.2%), recall (97.4%), and F1-score (98.3%), respectively. The Random Forest model demonstrated equally strong performance, achieving 99.8% recall, 99.0% precision, and 99.4% F1-score (Figure 7b). Importantly, the deep learning classifier achieved an 86% reduction in required manual verification (291,501 of 339,869 images). Meanwhile, our analysis revealed a 10% error rate in OSM's global wind turbine dataset. While this validates its reliability for macro-scale trend analysis, the findings underscore inherent limitations of data directly obtained from OSM for precision-critical wind energy applications.

**Figure 7.** Evaluation results of two models for wind turbine classification. **(a)** Precision, recall, and F1-score of the deep learning model. **(b)** Precision, recall, and F1-score of the traditional machine learning model.

# 3.2 Comparison with open-source datasets

To validate the accuracy of our wind turbine records, we cross-validated them against multiple authoritative geospatial datasets, including the 2020 global wind and solar dataset (Dunnett et al., 2020), along with official and research-based turbine inventories from the United States

(Rand et al., 2020), Italy (Smeraldo et al., 2020), and Germany (Manske et al., 2022). Our 2024 global inventory documents 379,595 wind turbines (**Table 1**), representing a tenfold expansion from the 2020 baseline of 33,514 turbines. Our wind turbine count closely aligns with Global Renewables Watch's 2024 total of ~375,000 wind turbines, showing a merely 1.2% variance. The consistency between our United States estimates (74,487 turbines) and official revealed records (75,781 turbines, 

**Figure 8.** Global onshore wind turbine installation records and spatial distribution. **(a)** Global onshore wind turbine by 2024. **(b)** Spatial distribution of wind turbine installation statistics by country. **(c)** Percentage ranking of wind turbines for top 20 countries.

# 3.4 Land use types and spatial distribution of global onshore wind turbines

Our global assessment quantifies a total impacted area of 242,940 km² of the wind turbines, which is estimated with 800-meter buffer around turbine locations (Dunnett et al., 2020). Among the affected areas, 80% of wind turbines located within cropland and grassland ecosystems (**Figure 9c**). Specifically, croplands represent the predominant land use at 42% (100,915 km²), followed by grasslands for 38% (93,028 km²), and forests for 12% (29,832 km²). These proportions, however, exhibit substantial variation across national boundaries (**Figure 9a, b**). China, the global leader in wind capacity, exhibits unique siting patterns with over 50% of turbines deployed in grasslands, followed by croplands (20%) and forests (15%). China

demonstrates a notably higher reliance on forested areas for wind turbine siting compared to global patterns, particularly in its southern provinces (Figure 9a), warranting careful ecological assessment (Enevoldsen, 2016). In contrast, the United States distributes roughly half (50%) of its wind turbines across croplands, supplemented by grassland deployments. Germany displays the most extreme geographic specialization, with over 90% of its turbines sited exclusively on agricultural lands. These pronounced regional variations in turbine siting patterns carry significant implications for both renewable energy development and landscape management policies.

**Figure 9.** Land use distribution of global onshore wind turbines.(a) Land use distribution of global onshore wind turbines. (b) Land use area statistics occupied by onshore wind turbines by country. (c) Percentages of difference land use deployed by onshore wind turbines.

## 3.5 Potential dataset applications

This open-access global onshore wind turbine dataset could establish a critical foundation for interdisciplinary research, facilitating integrated studies in energy infrastructure planning, ecological impact evaluation, and land use optimization. First, the geospatial wind turbine dataset enables rigorous biodiversity impact assessments, including wildlife disturbance patterns and habitat fragmentation analysis around wind energy installations (Bopucki and

302 Perzanowski, 2018; Mckay et al., 2024). Particularly, studies have demonstrated that turbine 303 blade rotation creates distinct mortality patterns across bird and bat species (Marques et al., 304 2020; Millon et al., 2018). Our precisely geolocated turbine records enable exact spatial 305 correlation between wind infrastructure and vulnerable species' high-activity areas, facilitating 306 data-driven assessments of avian and chiropteran collision risks. 307 Second, wind farm construction and associated infrastructure development induce significant 308 ecological disruptions through multiple pathways (Xia et al., 2025). Integrating high-precision 309 turbine locations with remote sensing data allows systematic evaluation of wind energy's 310 environmental footprint, including deforestation patterns (Enevoldsen, 2018), soil erosion (Ma 311 et al., 2023), and carbon sink loss (Gao et al., 2023). Our dataset provides a robust data 312 foundation for both evaluating the cumulative ecological impacts of existing wind farms and 313 optimizing future turbine siting to balance energy production with ecosystem conservation.

# 4 Data availability

- These open-access data resources could help promote transparent and just sustainable wind energy development, and enable detailed feature extraction and spatial analysis for future wind energy research. The global onshore wind turbine dataset is freely available from the Zenodo website at: https://doi.org/10.5281/zenodo.17217523 (Shujun et al., 2025).
- The dataset includes:
- A comprehensive global inventory of 379,595 onshore wind turbines in the format of a geospatial shapefile. The dataset includes geolocation coordinates for all wind turbines, along with corresponding nation (Field: 'Nation') and land use classification (Field: 'landtype') for each wind turbine.
- The dataset comprises 37,285 carefully annotated 256×256 pixel Google Earth image patches, containing both positive (wind turbine) and negative (background) samples, and is organized into folders with training (60%, 22,372 images) and testing sets (20%, 7,457 images) validating sets (20%, 7,456 images). The images could serve as foundational data for training deep learning models in wind turbine classification, segmentation, and detection tasks.
- The code file includes:

- A PyTorch-based ResNet-18 implementation for classifying onshore wind turbines in
   Google Earth images, including codes for model architecture and pre-trained weights.
- The GEE-based code for the Random Forest model, including sample point splitting (training/test sets) and model training.

This study introduces an advanced geospatial approach that integrates high-resolution Google

# 5 Discussion and conclusion

337 Earth images with multi-source satellite observations to construct a refined global inventory of 338 onshore wind turbines. Compared current datasets of available global onshore wind turbines, 339 our dataset more timely data that represents a tenfold expansion over the global wind turbine 340 inventories as of 2020. Importantly, in mapping methodology, compared to the new updating 341 framework of Global Renewables Watch, we propose a reproducible and straightforward 342 approach to identify renewable infrastructure, which can be applied in future studies and in 343 countries or regions with limited computational resources. The datasets and resulting 2024 344 global inventory documents 379,595 onshore wind turbines, serving as a critical resource for 345 renewable energy infrastructure planning and ecological impact studies. 346 The global analysis demonstrates significant spatial aggregation of wind turbines, with the densest concentrations occurring in northern mid-latitude zones, particularly high-density 347 348 concentrations in Europe, North America, and East Asia. This spatial concentration pattern 349 stems from factors including optimal wind resources (Davis et al., 2023; Liu et al., 2023), 350 supportive policy frameworks (Godby et al., 2025; Kumar et al., 2022; Liao, 2016), and 351 established energy infrastructure networks (Oró et al., 2015; Rochmińska, 2023) prevalent in 352 these mid-latitude zones. Notably, the global wind energy has developed across 242,940 km<sup>2</sup> 353 of land, with agricultural fields (42%) and grasslands (38%) hosting the majority (80%) of 354 turbine installations. This distribution reflects a strategic preference for siting turbines in 355 previously developed or ecologically low-sensitivity areas. However, the associated ecological 356 impacts, particularly habitat fragmentation and soil disturbance, require thorough 357 environmental evaluation and mitigation planning (Moore O'Leary et al., 2017). 358 Wind turbines primarily appear as point features in satellite images, presenting significant 359 challenges for automated large-scale detection (Zhai et al., 2024). These detection challenges

360 are further intensified by visually similar infrastructure, particularly high-voltage transmission 361 lines and isolated structures that mimic turbine signatures. Our proposed solution combines hybrid machine/deep learning architectures with systematic sampling approaches to enable 362 363 reliable turbine identification across diverse terrain types. Looking ahead, sustainable 364 renewable energy development, including wind, solar, and hydropower, requires continuous 365 innovation and open geospatial data to enhance planning transparency and governance. Overall, 366 our framework offers a novel approach and solution for cost-effective, timely updates of global onshore wind turbine data. 367 368 Author contributions. SL and PW designed the study and wrote the original manuscript. SL designed the methods and carried out the experiments and validation. JQ and YS edited and 369 370 revised the paper. 371 Competing interests. The contact author has declared that none of the authors has any 372 competing interests. 373 Disclaimer. Publisher's note: Copernicus Publications remains neutral with regard to 374 jurisdictional claims in published maps and institutional affiliations. 375 Acknowledgements. The authors are grateful to the ESA's Copernicus program for providing 376 free access to the Sentinel-1/2 data and Google Earth Engine platform for preprocessing and 377 making the data accessible. We also thank the OpenStreetMap for providing global onshore 378 wind turbine locations and the land polygon data. 379 Financial support. This work was supported by National Key Research and Development 380 Program (Grant No.2023YFC3904500). 381 References 382 Bopucki, R. and Perzanowski, K.: Effects of wind turbines on spatial distribution of the 383 European hamster, Ecol. Indic., 84, 433-436, 2018. 384 Calvert, K., Pearce, J. M., and Mabee, W. E.: Toward renewable energy geo-information 385 infrastructures: applications of geoscience and remote sensing that build institutional 386 capacity, Renewable and Sustainable Energy Reviews, 18, 416-429, 2013. 387 Cerri, J., Costantino, C., De Rosa, D., Banič, D. A., Urgeghe, G., Fozzi, I., Echeverria, J., Aresu, M., and Berlinguer, F.: Widely used datasets of wind energy infrastructures can seriously 388 389 underestimate onshore turbines in the mediterranean, Biol. Conserv., 300, 110870, 2024. 390 Congalton, R. G.: A review of assessing the accuracy of classifications of remotely sensed data, 391 Remote Sens. Environ., 37, 35-46, 1991.

- Land resources for wind energy development requires regionalized characterizations, Environ. Sci. Technol., 58, 5014-5023, 2024.
- Davis, N. N., Badger, J., Hahmann, A. N., Hansen, B. O., Mortensen, N. G., Kelly, M., Larsén,
- X. G., Olsen, B. T., Floors, R., and Lizcano, G.: The global wind atlas: a high-resolution
- dataset of climatologies and associated web-based application, Bull. Amer. Meteorol. Soc.,
   104, E1507-E1525, 2023.
- Dunnett, S., Sorichetta, A., Taylor, G., and Eigenbrod, F.: Harmonised global datasets of wind and solar farm locations and power, Sci. Data, 7, 130, 2020.
- Effenberger, N. and Ludwig, N.: A collection and categorization of open source wind and wind power datasets, Wind Energy, 25, 1659-1683, 2022.
- Enevoldsen, P.: Onshore wind energy in northern European forests: reviewing the risks, Renewable and Sustainable Energy Reviews, 60, 1251-1262, 2016.
- Enevoldsen, P.: A socio-technical framework for examining the consequences of deforestation:
  a case study of wind project development in northern Europe, Energy Policy, 115, 138147, 2018.
- Gao, L., Wu, Q., Qiu, J., Mei, Y., Yao, Y., Meng, L., and Liu, P.: The impact of wind energyon plant biomass production in China, Sci. Rep., 13, 22366, 2023.
- Godby, R., Cook, B., Holland, M., and Kjorstad, T.: State incentives: impact on wind energy
   costs and policy development, Renewable and Sustainable Energy Reviews, 215, 115572,
   2025.
- Goutte, C. and Gaussier, E.: A probabilistic interpretation of precision, recall and f-score, with implication for evaluation, in: European conference on information retrieval, Springer, 345-359, 2005.
- He, K., Zhang, X., Ren, S., and Sun, J.: Deep residual learning for image recognition, in:
   Proceedings of the IEEE conference on computer vision and pattern recognition, IEEE,
   Las Vegas, NV, USA, 770-778, 2016.
- Hoeser, T., Feuerstein, S., and Kuenzer, C.: DeepOWT: a global offshore wind turbine data set
   derived with deep learning from sentinel-1 data, Earth System Science Data Discussions,
   2022, 1-26, 2022.
- Huang, S., Tang, L., Hupy, J. P., Wang, Y., and Shao, G.: A commentary review on the use of
   normalized difference vegetation index (NDVI) in the era of popular remote sensing, J.
   For. Res., 32, 1-6, 2021.
- Karra, K., Kontgis, C., Statman-Weil, Z., Mazzariello, J. C., Mathis, M., and Brumby, S. P.:
   Global land use/land cover with sentinel 2 and deep learning, in: 2021 IEEE international
   geoscience and remote sensing symposium IGARSS, IEEE, Brussels, Belgium, 4704 4707, 2021.
- Kati, V., Kassara, C., Vrontisi, Z., and Moustakas, A.: The biodiversity-wind energy-land use
   nexus in a global biodiversity hotspot, Sci. Total Environ., 768, 144471, 2021.
- Kruitwagen, L., Story, K. T., Friedrich, J., Byers, L., Skillman, S., and Hepburn, C.: A global
   inventory of photovoltaic solar energy generating units, Nature, 598, 604-610, 2021.
- Kumar, A., Pal, D., Kar, S. K., Mishra, S. K., and Bansal, R.: An overview of wind energy
   development and policy initiatives in India, Clean Technol. Environ. Policy, 24, 1337 1358, 2022.
- Liao, Z.: The evolution of wind energy policies in China (1995 2014): an analysis based on

- policy instruments, Renewable and Sustainable Energy Reviews, 56, 464-472, 2016.
- Liu, Y., Feng, S., Qian, Y., Huang, H., and Berg, L. K.: How do north American weather 439 regimes drive wind energy at the sub-seasonal to seasonal timescales? Npj Clim. Atmos. 440 Sci., 6, 100, 2023.
- Ma, B., Yang, J., Chen, X., Zhang, L., and Zeng, W.: Revealing the ecological impact of low-speed mountain wind power on vegetation and soil erosion in south China: a case study of
   a typical wind farm in Yunnan, J. Clean. Prod., 419, 138020, 2023.
- Manske, D., Grosch, L., Schmiedt, J., Mittelstädt, N., and Thrän, D.: Geo-locations and system
   data of renewable energy installations in Germany, Data, 7, 128, 2022.
- Marques, A. T., Santos, C. D., Hanssen, F., Muñoz, A. R., Onrubia, A., Wikelski, M., Moreira,
   F., Palmeirim, J. M., and Silva, J. P.: Wind turbines cause functional habitat loss for
   migratory soaring birds, J. Anim. Ecol., 89, 93-103, 2020.
- Mckay, R. A., Johns, S. E., Bischof, R., Matthews, F., van der Kooij, J., Yoh, N., and Eldegard,
   K.: Wind energy development can lead to guild specific habitat loss in boreal forest bats,
   Wildlife Biol., 2024, e1168, 2024.
- Mckenna, R., Lilliestam, J., Heinrichs, H. U., Weinand, J., Schmidt, J., Staffell, I., Hahmann,
   A. N., Burgherr, P., Burdack, A., and Bucha, M.: System impacts of wind energy
   developments: key research challenges and opportunities, Joule, 9, 2025.
- Millon, L., Colin, C., Brescia, F., and Kerbiriou, C.: Wind turbines impact bat activity, leading to high losses of habitat use in a biodiversity hotspot, Ecol. Eng., 112, 51-54, 2018.
- Mishra, M., Desul, S., Santos, C. A. G., Mishra, S. K., Kamal, A. H. M., Goswami, S., Kalumba,
   A. M., Biswal, R., Da Silva, R. M., and Dos Santos, C. A. C.: A bibliometric analysis of
   sustainable development goals (SDGs): a review of progress, challenges, and
   opportunities, Environment, Development and Sustainability, 26, 11101-11143, 2024.
- Moore O'Leary, K. A., Hernandez, R. R., Johnston, D. S., Abella, S. R., Tanner, K. E., Swanson,
   A. C., Kreitler, J., and Lovich, J. E.: Sustainability of utility scale solar energy critical
   ecological concepts, Front. Ecol. Environ., 15, 385-394, 2017.
- Muller, E., Gremmo, S., Houtin-Mongrolle, F., Duboc, B., and Bénard, P.: Field-data-based
   validation of an aero-servo-elastic solver for high-fidelity large-eddy simulations of
   industrial wind turbines, Wind Energy Sci., 9, 25-48, 2024.
- Oró, E., Depoorter, V., Garcia, A., and Salom, J.: Energy efficiency and renewable energy
   integration in data centres. Strategies and modelling review, Renewable and Sustainable
   Energy Reviews, 42, 429-445, 2015.
- Raimi, D., Zhu, Y., Newell, R. G., Prest, B. C., and Bergman, A.: Global energy outlook 2023:
  sowing the seeds of an energy transition, Resour. Future, 1, 1-44, 2023.
- Rand, J. T., Kramer, L. A., Garrity, C. P., Hoen, B. D., Diffendorfer, J. E., Hunt, H. E., and 473 Spears, M.: A continuously updated, geospatially rectified database of utility-scale wind 474 turbines in the United States, Sci. Data, 7, 15, 2020.
- Rigatti, S. J.: Random Forest, Journal of Insurance Medicine, 47, 31-39, 2017.
- Rinne, E., Holttinen, H., Kiviluoma, J., and Rissanen, S.: Effects of turbine technology and land 477 use on wind power resource potential, Nat. Energy, 3, 494-500, 2018.
- Robinson, C., Ortiz, A., Kim, A., Dodhia, R., Zolli, A., Nagaraju, S. K., Oakleaf, J., Kiesecker,
- J., and Ferres, J. M. L.: Global renewables watch: a temporal dataset of solar and wind energy derived from satellite imagery, Arxiv Preprint Arxiv:2503.14860, 2025.

# https://doi.org/10.5194/essd-2025-512 Preprint. Discussion started: 29 October 2025 © Author(s) 2025. CC BY 4.0 License.

- Rochmińska, A.: Wind energy infrastructure and socio-spatial conflicts, Energies, 16, 1032, 2023.
- Roddis, P., Carver, S., Dallimer, M., Norman, P., and Ziv, G.: The role of community acceptance in planning outcomes for onshore wind and solar farms: an energy justice analysis, Appl. Energy, 226, 353-364, 2018.
- Shujun, L., Jianchuan, Q., Yongze, S., and Wang, P.: Mapping global onshore wind turbines using multi-source remote sensing images and hybrid learning approaches, 2025. https://doi.org/10.5281/zenodo.16759861.
- Smeraldo, S., Bosso, L., Fraissinet, M., Bordignon, L., Brunelli, M., Ancillotto, L., and Russo,
   D.: Modelling risks posed by wind turbines and power lines to soaring birds: the black
   stork (ciconia nigra) in italy as a case study, Biodivers. Conserv., 29, 1959-1976, 2020.
- Tavakkoli, S., Macknick, J., Heath, G. A., and Jordaan, S. M.: Spatiotemporal energy 493 infrastructure datasets for the United States: a review, Renewable and Sustainable Energy 494 Reviews, 152, 111616, 2021.
- Xia, Z., Li, Y., Guo, S., Zhang, X., Pan, X., Fang, H., Chen, R., and Du, P.: Assessment of
   forest disturbance and soil erosion in wind farm project using satellite observations,
   Resources, Conservation and Recycling, 212, 107934, 2025.
- Zha, Y., Gao, J., and Ni, S.: Use of normalized difference built-up index in automatically
   mapping urban areas from tm imagery, Int. J. Remote Sens., 24, 583-594, 2003.
- Zhai, Y., Chen, X., Cao, X., and Cui, X.: Identifying wind turbines from multiresolution and
   multibackground remote sensing imagery, Int. J. Appl. Earth Obs. Geoinf., 126, 103613,
   2024.
- Zhang, T., Tian, B., Sengupta, D., Zhang, L., and Si, Y.: Global offshore wind turbine dataset,
   Sci. Data, 8, 191, 2021.

505506