# Peer review of "Mapping global onshore wind turbines using multi-source remote"

_Earth System Science Data, 2025_

## Referee Comment (RC1)

**Spelling and Grammar Issues**

ZZ 45–46
Original: "The codes and dataset ... is available."
Correct: "The code and dataset ... are available."

ZZ 83
Original: "there is a geospatial wind turbine dataset for 2020 is introduced."
Correct: "A geospatial wind turbine dataset for 2020 was introduced."

**Unclear Description: OSM Query**

The query string in line 129 (["generator: source"="wind"]) appears syntactically incorrect.

OpenStreetMap commonly uses:

- generator:source=wind

- or power=generator combined with generator:source=wind

The manuscript should clearly state the exact Overpass query used.

**Major Comment: Missing Methodological Detail on OSM Extraction**

- The manuscript does not specify whether nodes, ways, or relations were extracted.
- It is unclear whether both power=generator and generator:source=wind were used.
- The handling of offshore turbines is not described.
- No link to the exact Overpass script is provided.
- There is no information on whether wind farms, meteorological masts, or power-line infrastructure were filtered out.
- These omissions significantly limit reproducibility and must be clarified.

**Major Comment: Unexplained OSM Error Rate**

The manuscript states a "10% error rate in OSM's global wind turbine dataset" but provides no methodological explanation.

Missing information:

- How was this error rate calculated? What was the validation procedure?

- Was the error rate spatially or regionally variable?

- Were commission and omission errors distinguished?

- Was the result compared to existing studies with likewise approaches, e.g. https://www.mdpi.com/2220-9964/14/6/232

**Major Comment: Insufficient Documentation of Random Forest Sampling**

Missing details include:

- From which land cover classes were negative samples drawn?

- Was spatial autocorrelation considered?

- How was it ensured that negative samples were not within 30 m of existing turbines?

- Was the global distribution of positive and negative samples balanced?

The Random Forest sampling workflow requires a clear methodological description.

**Minor Comment: Sentinel-1/2 Features and Missing GEE Scripts**
The manuscript lists processing steps, but details are missing, included:

- Exact satellite collections used.

- Time span and temporal compositing strategy.

- Cloud masking method (e.g., QA60 for Sentinel-2).

- Preprocessing steps such as resampling, mosaicking, and normalisation.

Referencing Zenodo alone is insufficient; the core processing steps must appear in the manuscript.